# Association between Physical Activity Habits with Cardiometabolic Variables, Body Composition, and Physical Performance in Chilean Older Women

**DOI:** 10.3390/ijerph20176688

**Published:** 2023-08-31

**Authors:** Jordan Hernandez-Martinez, Camila González-Castillo, Tomás Herrera-Valenzuela, Cristopher Muñoz-Vásquez, Braulio Henrique Magnani Branco, Pablo Valdés-Badilla

**Affiliations:** 1Department of Physical Activity Sciences, Universidad de Los Lagos, Osorno 5290000, Chile; jordan.hernandez@ulagos.cl; 2Programa de Investigación en Deporte, Sociedad y Buen Vivir, Universidad de los Lagos, Osorno 5290000, Chile; 3Department of Health, Universidad de los Lagos, Osorno 5290000, Chile; camila.gonzalez1@ulagos.cl; 4Department of Physical Activity, Sports and Health Sciences, Faculty of Medical Sciences, Universidad de Santiago de Chile (USACH), Santiago 8370003, Chile; tomas.herrera@usach.cl; 5Programa de Prevención y Rehabilitación Cardiovascular, CESFAM Dr. Juan Carlos Baeza Bustos, Departamento de Salud San Clemente, San Clemente 3520000, Chile; cristophermunoz@saludsanclemente.cl; 6Programa de Magíster en Ciencias de la Actividad Física, Facultad de Ciencias de la Educación, Universidad Católica del Maule, Talca 3480094, Chile; 7Postgraduate Program in Health Promotion, Cesumar University, Maringá 87050-390, PR, Brazil; braulio.branco@unicesumar.edu.br; 8Department of Physical Activity Sciences, Faculty of Education Sciences, Universidad Católica del Maule, Talca 3530000, Chile; 9Sports Coach Career, School of Education, Universidad Viña del Mar, Viña del Mar 2520000, Chile

**Keywords:** exercise, obesity, physiology, anthropometry, physical fitness, older adults, aging

## Abstract

This study aimed to associate physical activity habits with cardiometabolic variables (blood pressure, fasting glucose, HDL cholesterol, and triglycerides), body composition (body fat percentage and fat-free mass), and physical performance (handgrip strength (HGS), timed up-and-go (TUG), and walking speed) in Chilean older women. An analytical cross-sectional study analyzed 179 older women with a mean age of 75.4 years distributed into physically inactive (PI) older women (*n* = 74) and physically active (PA) older women (*n* = 105). A logistic regression showed that PI older women presented an increased risk of hyperglycemia (OR = 4.70; *p* = 0.000), high blood pressure (OR = 3.83; *p* = 0.000), low HDL cholesterol levels (OR = 2.13; *p* = 0.03), hypertriglyceridemia (OR = 2.54; *p* = 0.01), excess body fat percentage (OR = 4.33; *p* = 0.000), low fat-free mass (OR = 2.22; *p* = 0.02), low HGS in their dominant hand (OR = 3.37; *p* = 0.001) and non-dominant hand (OR = 3.60; *p* = 0.0001), and poor performance in TUG (OR = 5.60; *p* = 0.000) and walking speed (OR = 5.52; *p* = 0.000). In conclusion, physical inactivity was associated with increased cardiometabolic risk, excess body fat percentage, lower fat-free mass, and poorer physical performance in Chilean older women. At the same time, PA older women showed a lower cardiometabolic risk, better body composition, and better physical performance than PI older women.

## 1. Introduction

Low physical activity levels negatively affect health status; the worldwide extent of physical inactivity has reached 31% among older people [1]. In Chile, about 90% are sedentary with low levels of physical activity, according to the “Servicio Nacional del Adulto Mayor” [2]. The COVID-19 pandemic was a factor that led to self-isolation, drastic changes in lifestyles and social behavior, and decreases in physical activity habits in older people, reaching 53% [3]. A higher percentage of low physical activity levels of 61.7% were found in Chilean women during the COVID-19 pandemic [4]. Low physical activity levels have been associated with increased cardiometabolic risk in older people [5,6], an increase in body fat percentage, along with a decrease in fat-free mass by 13% to 16% [7], together with alterations in physical performance [1], as well as muscle strength, postural balance, and walking speed [7], and has been estimated to cost $53.8 billion in direct healthcare expenses annually in the world population [1].

However, leading an active and healthy lifestyle while aging through regular physical activity of moderate intensity (≥150 to 300 min per week) and/or vigorous intensity (≥75 to 150 min per week) or their combination [8] leads to lower cardiometabolic risk [9], lower body fat percentage and higher lean mass [10], and improved physical performance in older people [11], which together can affect their autonomy and health-related quality of life [12]. In a systematic review with a meta-analysis conducted by Kazeminia et al. [13], a lower risk of hypertension was reported with moderate (OR = 0.89; 95%CI = 0.80 to 0.99; *p* < 0.05) and vigorous (OR = 0.82; 95%CI = 0.73 to 0.93; *p* ≤ 0.01) physical activity in older people. A study by Lee et al. [14] observed an association (OR = 0.65; 95%CI = 0.45 to 0.94; *p* < 0.05) between high levels of vigorous physical activity and a lower risk of cardiometabolic syndrome in Korean older people. Similarly, Xu et al. [15] found an association between high physical activity levels and high levels of high-density lipoproteins (OR = 0.39; 95%CI = 0.18 to 0.84; *p* = 0.04), a lower risk of arterial hypertension (OR = 0.39; 95%CI = 0.20 to 0.77; *p* = 0.02), and hyperglycemia (OR = 0.34; 95%CI = 0.15 to 0.78; *p* = 0.02) in older people of the United States of America. In another study by Jaclyn, Emily, Heather, and Monica [10], it was observed that high levels of moderate to vigorous physical activity led to a lower body fat percentage (OR = −0.1; 95%CI = −0.1 to 0.0; *p* < 0.001) and greater lean mass (OR = 0.1, 95%CI = 0.0 to 0.1, *p* = 0.03) in Canadian older people. Regarding physical performance, de Araújo Amaral et al. [16] reported that higher levels of moderate to vigorous physical activity led to higher handgrip strength (HGS) (OR = 1.75; 95%CI = 1.08 to 2.84; *p* = 0.007) in Brazilian older people. However, in a study by Chen et al. [17], it was reported that physically inactive (PI) older people had a poorer performance in timed up-and-go (TUG; *p* < 0.01) and walking ability (*p* < 0.01) compared to the physically active (PA).

Although reputable reports have indicated the association between physical activity habits and better cardiometabolic variables, body composition, and physical performance in older people [10,13,14,15,16,17], it is still necessary to understand the physical activity habits post-COVID-19 pandemic and their relationship with health status in the local population. Therefore, the main aim of this study was to associate physical activity habits (PA vs. PI) with cardiometabolic variables (blood pressure, fasting glucose, HDL cholesterol, and triglycerides), body composition (body fat percentage and fat-free mass), and physical performance (HGS, TUG, and walking speed) in Chilean older women. Secondarily, we aimed to compare PA older women to PI older women in terms of cardiometabolic variables, body composition, and physical performance. Physical inactivity is hypothesized to be a risk factor for increased cardiometabolic risk, poor body composition, and physical performance [13,14,15,16]. Secondly, PA older women have better cardiometabolic variables, body composition, and physical performance than PI older women [18,19,20].

## 2. Material and Methods

### 2.1. Study Design

This study presents an analytical cross-sectional design [21]. Older women belonging to social programs without physical activity and physical activity groups focused on older people were invited to participate, distributing them according to physical activity habits into PI older women (*n* = 74) and PA older women (*n* = 105). The assessments of blood pressure, fasting glucose, HDL cholesterol, triglycerides, body fat percentage, fat-free mass, HGS, TUG, and walking speed were made. These assessments were carried out in a laboratory with controlled conditions in November 2022 in the morning on two days with a 72 h recovery period between each day. On day 1, fasting blood pressure measurements (systolic and diastolic), fasting glucose, HDL cholesterol, triglycerides, body fat percentage, and fat-free mass were taken. On day 2, HGS, TUG, and walking speed measurements were taken.

### 2.2. Participants

One hundred seventy-nine Chilean older women (75.4 ± 4.5 years old) were selected. The sample size calculation was made from 310 older women enrolled in all groups of older people in Osorno, Chile, who were PA and PI. The calculation estimate was 170 participants. This research was carried out as recommended in a previous study by Fang et al. [22]. For this calculation, a confidence level of 95% and a margin of error of 5% were used. These analyses were performed using GPower software (version 3.1.9.6, Franz Faul, Universiät Kiel, Kiel, Germany). Older women who met international recommendations for moderate (≥150 to 300 min) or vigorous (≥75 to 150 min) physical activity per week were considered PA [23]; those who did not meet these recommendations were considered PI (see Table 1). The inclusion criteria were: (i) women over 60 years of age; (ii) physical condition compatible with the practice of physical activity; (iii) the ability to understand and follow instructions in a contextualized manner employing simple commands. The following were excluded: (i) older women who presented any cardiovascular, respiratory pathology, or musculoskeletal injury that prevented them from practicing physical activity; (ii) those with moderate or severe cognitive impairment (≥15) assessed by the abbreviated Mini-Mental State Examination [24]; (iii) present functional dependence for activities of daily living (moderate = 40 to 55 points, severe = 20 to 35 points, and/or total =< 20 points) measured through the Barthel Index according to the Ministerio de Salud [25]; (iv) those who presented any sequelae due to COVID-19 at the neurological, respiratory, and/or cardiovascular level that prevented them from carrying out the physical performance assessments. Figure 1 shows the sample selection process.

All participants had to accept the criteria for using and handling the data by signing an informed consent authorizing the use of the information for scientific purposes. The research protocol was reviewed and approved by the Scientific Ethics Committee of the Universidad Autónoma de Chile (approval number: 06-2016) and was developed following the guidelines of the Helsinki Declaration regarding research involving human subjects.

### 2.3. Cardiometabolic Variables

The outcomes were cardiometabolic variables, including systolic blood pressure (SBP), diastolic blood pressure (DBP), fasting glucose, HDL cholesterol, and triglycerides [5]. Procedures to measure blood pressure were carried out as proposed by Reddy et al. [26]. The older women sat in a comfortable chair for 15 min in a quiet room. After this period, a cuff was placed on the midpoint of the upper left arm (heart level). An automatic blood pressure monitor (model HEM-7121, Omron, Osaka, Japan) was used to measure SBP and DBP. During blood pressure recording, the participants remained relaxed in the seated position, feet parallel to shoulder width, both forearm and hands on the table, hands supinated, and backs against the chair, without moving or talking. The older women did not have access to the blood pressure values during the measurement. The assessments lasted approximately 80 s and were performed three times with a one-minute rest between assessments. The mean of each person’s measurements was used in the final analysis.

Fasting blood samples were collected by venous puncture, in which a sample is extracted and processed to achieve HDL cholesterol and triglyceride parameters. The fasting glucose of the day was detected by taking a sample of capillary blood extracted from a small puncture of the nail bed of each older woman. The sample is deposited in a reactive tape connected to a glucometer, yielding the fasting glucose value that the patient presents at that moment. All measurements to determine cardiometabolic risk were performed by a nurse in a laboratory with optimal conditions for these measurements. As proposed by Lee, Kim, and Jeon [14], an older woman with hypertriglyceridemia must have a triglyceride concentration of ≥150 mg/dL, a low HDL cholesterol concentration of <50 mg/dL, a hyperglycemia concentration of ≥100 mg/dL, and a high blood pressure of SBP ≥ 130 and DBP ≥ 85 mmHg.

### 2.4. Body Composition

Bipedal height was measured using a stadiometer (Seca model 220, SECA, Hamburg, Germany; accuracy to 0.1 cm), and body weight was calculated using a mechanical scale (Scale-tronix, Chicago, IL, USA; accuracy to 0.1 kg) while wearing the barest minimum of clothing. The body fat percentage and fat-free mass were calculated using tetrapolar bioimpedance (InBody 570^®^, Seoul, Republic of Korea) with eight tactile point electrodes. The International Society for the Advances in Kinanthropometry (ISAK) recommendations were followed for each measurement [27]. These characteristics are reported in Table 1.

### 2.5. Physical Performance

#### 2.5.1. Handgrip Strength (HGS)

According to earlier suggestions, HGS was used [28]. It was discovered that a sedentary posture, with the spine aligned, the shoulder in neutral, the elbow flexed at 90 degrees to the side of the body, and the forearm and wrist in neutral was ideal for the test. The test used a handheld dynamometer (Jamar^®^, PLUS+, Sammons Preston, Patterson Medical, Warrenville, IL, USA). According to the size of the hand, the dynamometer’s position was chosen to allow for a secure grip on the instrument while maintaining proper closure of the metacarpal phalangeal and interphalangeal joints in the first position, which favors contact between the first phalanx of the index finger and the thumb. With a 120 s rest, each participant made three attempts with each hand. A blinded assessor recorded the maximum value of the three efforts performed by each participant for each hand.

#### 2.5.2. Timed Up-and-Go Test (TUG)

Following earlier suggestions, the TUG test was conducted [29]. The person must get out of an arm-supported chair, cross a three-meter aisle, turn around, and return to the chair. They must perform three trials and record the best one in seconds. A pair of evaluators measured the time using single-beam photocells (Brower Timing System, Draper, UT, USA), and the best of three trials was utilized for statistical analysis.

#### 2.5.3. Walking Speed

Single-beam photocells (Brower Timing System) were used to measure walking speed. Under earlier guidelines, the test was conducted [30]. The nearest 0.01 s to the photocells was used to measure time. Participants made three submaximal efforts for test familiarization while moving down a four-meter hallway. They then made three maximum trials, with at least a minute of rest between each. For the three trials, participants were instructed to walk as quickly as they could; the best of the three trials was used for statistical analysis.

### 2.6. Statistical Analysis

Data were analyzed with SPSS 25.0 statistical software (SPSS 25.0 for Windows, SPSS Inc., Chicago, IL, USA). Values were reported as mean ± standard deviation. The Kolmogorov–Smirnov test was used to determine the normality of the data, while Levene’s test was used to determine the homogeneity of variance. A normal distribution was observed for all data. A student’s *t*-test was performed for independent samples to compare the differences in physical activity habits (PA older women vs. PI older women). In comparison, factors of cardiometabolic variables (blood pressure, fasting glucose, HDL cholesterol, and triglycerides), body composition (body fat percentage and fat-free mass), and physical performance (HGS, TUG, and walking speed) were associated with physical activity habits (PA and PI), identified by logistic regression. These results were presented as odds ratios (OR) with their respective 95% confidence intervals (95%CI) to present the magnitude of the association. The level of significance was defined as *p* < 0.05.

## 3. Results

All reported means ± standard deviations for cardiometabolic variables, body composition, and physical performance are presented in Table 2.

Among the cardiometabolic variables, when comparing PA older women vs. PI older women, significant differences were reported in favor of PA older women for fasting glucose (F = 3.22; *p* = 0.000), SBP (F = 2.90; *p* = 0.000), DBP (F = 1.30; *p* = 0.000), HDL cholesterol (F = 1.25; *p* = 0.000), and triglycerides (F = 2.79; *p* = 0.000). These results are presented in Figure 2.

Similar to body composition and physical performance when comparing PA older women vs. PI older women, significant differences were reported in favor of PA older women for body fat percentage (F = 1.64; *p* = 0.000), fat-free mass (F = 2.95; *p* < 0.001), HGS in their dominant hand (F = 5.55; *p* = 0.000) and non-dominant hand (F = 11.2; *p* < 0.001), TUG (F = 1.05; *p* = 0.000), and walking speed (F = 2.78; *p* < 0.01). These results are presented in Figure 3.

In the logistic regression analyses of physical activity habits, it was reported that PI older women presented an increased risk in the following cardiometabolic variables: 4.7-fold for hyperglycemia (OR = 4.70; 95%CI = 2.16 to 10.9; *p* = 0.000), 3.8-fold for high blood pressure (OR = 3.83; 95%CI = 3.83 to 25.4; *p* = 0.000), 2.13-fold for low HDL cholesterol levels (OR = 2.13; 95%CI = 1.04 to 4.37; *p* = 0.03), and 2.5-fold for hypertriglyceridemia (OR = 2.54; 95%CI = 1.23 to 5.23; *p* = 0.01). Similarly, in terms of body composition, it was found that PI older women had a 4.3-fold greater risk of having an excess body fat percentage (OR = 4.33; 95%CI = 2.00 to 9.37; *p* = 0.000) and a 2.2-fold greater risk of having a low fat-free mass (OR = 2.22; 95%CI = 1.08 to 4.58; *p* = 0.02). In terms of physical performance, it was detected that PI older women had a 3.3-fold greater risk of having a low HGS in their dominant hand (OR = 3.37; 95%CI = 1.60 to 7.11; *p* = 0.001), a 3.6-fold greater risk in terms of HGS in their non-dominant hand (OR = 3.60; 95%CI = 1.89 to 6.84; *p* = 0.0001), along with a 2.44-fold poorer performance in terms of TUG (OR = 5.60; 95%CI = 2.44 to 12.6; *p* = 0.000) and a 5.5-fold slower walking speed (OR = 5.52; 95%CI = 2.46 to 12.3; *p* = 0.000). These results are presented in Figure 4. No significant associations were reported between the group of physically active older women and the analyzed variables.

## 4. Discussion

The main aim of this study was to associate physical activity habits with cardiometabolic variables, body composition, and physical performance in Chilean older women, and secondly, to compare PA older women with PI older women in terms of the analyzed variables. The main findings in the present study showed that physical inactivity was associated with increased fasting glucose, hypertriglyceridemia, high blood pressure, low HDL cholesterol levels, excess body fat percentage, a lower fat-free mass, a poorer HGS in dominant and non-dominant hands, TUG, and walking speed in older women. In addition, it was found that PA older women had a lower cardiometabolic risk, lower body fat percentage, higher fat-free mass, greater HGS in dominant and non-dominant hands, and better TUG and walking speed performance than PI older women. Therefore, the hypotheses were confirmed.

Similar to that reported by Li, White, O’Shields, McLain, and Merchant [19] in PI Chinese older people, an association was found between an increased risk of high cholesterol (OR = 6.89; 95%CI = 4.70 to 9.07; *p* < 0.01) and low HDL cholesterol levels (OR = 2.47; 95%CI = 1.67 to 3.27; *p* < 0.05). In the present study, it was reported that PI older women have a high cardiometabolic risk associated with hyperglycemia (OR = 4.70; 95%CI = 2.16 to 10.9; *p* = 0.000), high blood pressure (OR = 3.83; 95%CI = 3.83–25.4; *p* = 0.000), low HDL cholesterol levels (OR = 2.13; 95%CI = 1.04 to 4.37; *p* = 0.03), and hypertriglyceridemia (OR = 2.54; 95%CI = 1.23 to 5.23; *p* = 0.01). In this regard, Bowden Davies et al. [31] stated that physical inactivity in older people increases intrahepatic lipid buildup and systemic insulin resistance. However, one result reported in our study was that PA older women presented lower levels of fasting glucose (*p* = 0.000), SBP (*p* = 0.000), DBP (*p* = 0.001), and triglycerides (*p* = 0.004), along with higher concentrations of HDL cholesterol (*p* = 0.006) compared to PI older women. Amidei et al. [32] reported that a PA lifestyle has been associated with a lower cardiometabolic (OR = 0.74; 95%CI = 0.58 to 0.94; *p* = 0.03) and mortality (OR = 0.81; 95%CI = 0.72 to 0.92; *p* = 0.005) risk in older Italian people. Similar results were reported by Kazeminia, Daneshkhah, Jalali, Vaisi-Raygani, Salari, and Mohammadi [13] showing a lower risk of arterial hypertension following moderate (OR = 0.89; 95%CI = 0.80 to 0.99; *p* < 0.05) and vigorous (OR = 0.82; 95%CI = 0.73 to 0.93; *p* ≤ 0.01) physical activity in older people. On the other hand, regular physical activity stimulates the activation of AMP-activated protein kinase and glucose uptake in skeletal muscle mass; as a result, insulin sensitivity is preserved, and less glucose is diverted to metabolically unfavorable depots. At the cardiovascular level, the practice of physical activity increases the thickness of the myocardium, promotes the improvement of cardiac function, and improves the internal diameter and elasticity of the coronary arteries, thus improving the body’s cardiac function and reducing the hypertension risk [33].

Another result reported in the present study was the increased risk of an excess body fat percentage among PI older women (OR = 4.33; 95%CI = 2.00 to 9.37; *p* = 0.000), together with decreased fat-free mass (OR = 2.22; 95%CI = 1.08 to 4.58; *p* = 0.02). Similarly, in a systematic review with a meta-analysis, Silveira, Mendonça, Delpino, Elias Souza, Pereira de Souza Rosa, de Oliveira, and Noll [18] reported a significant association (OR = 1.52; 95%CI = 1.23 to 1.87; *p* < 0.01) between an increased obesity risk and PI older people. Similarly, Huh, Lee, and Son [20] reported a significant association (OR = 1.50; 95%CI = 1.20 to 2.20; *p* < 0.05) between physical inactivity and a low fat-free mass in Korean older women. Both obesity and low muscle mass can lead to adverse health effects in older people [34]. For example, obesity has been associated with increased cardiometabolic risk, more significant functional limitations, disability, and a poorer health-related quality of life in older people [35]. However, leading a PI lifestyle is related to increased anabolic endurance, increased muscle protein breakdown, and decreased muscle protein synthesis, leading to reduced satellite cell activation, thereby affecting muscle regeneration and leading to muscle atrophy in older people [31]. However, another result reported in our study was that physical activity led to a lower body fat percentage (*p* = 0.000) and a higher fat-free mass (*p* = 0.001) compared to PI older women. These results are similar to those reported by Jaclyn, Emily, Heather and Monica [10] in PA Canadian older people who had a lower body fat percentage (OR = −0.1, 95%CI = −0.1 to 0.0, *p* < 0.001), and higher fat-free mass (OR = 0.1, 95%CI = 0.0 to 0.1, *p* = 0.03) compared to PI older people. Regular physical activity stimulates mitochondrial biogenesis, leading to an increased percentage of body fat oxidation in older people [36]. However, in terms of muscle mass, physical activity in older people increases the anabolic sensitivity of the muscle, helping the accelerated reduction of fat-free mass and leading to better muscle quality [31].

It was also reported in the present study that PI older women presented a worse HGS in their dominant hand (OR = 3.37; 95%CI = 1.60 to 7.11; *p* = 0.001), HGS in their non-dominant hand (OR = 3.60; 95%CI = 1.89 to 6.84; *p* = 0.0001), TUG (OR = 5.60; 95%CI = 2.44 to 12.6; *p* = 0.000), and walking speed (OR = 5.52; 95%CI = 2.46 to 12.3; *p* = 0.000). A decreased HGS has been associated with increased mortality risk (hazard ratio = 1.39; 95%CI = 1.13 to 1.71; *p* < 0.001) in Chilean older people [37], while older people with a poor performance in terms of TUG (*p* < 0.001) and walking speed (*p* < 0.01) have an increased fall risk and frailty [38]. However, other results reported in the present study showed significant differences (*p* = 0.000) in favor of PA older women compared to PI older women in terms of HGS in dominant and non-dominant hands, TUG, and walking speed. Results similar to those obtained by de Araújo Amaral, Amaral, Monteiro, de Vasconcellos, and Portela [16] reported that higher physical activity led to a higher HGS in the dominant hands (OR = 1.75; 95%CI = 1.08 to 2.84, *p* = 0.007) of Brazilian older people. Both gains in HGS, TUG, and walking speed performance have been associated with increased autonomy and a better health-related quality of life in older people [12,37,38]. Therefore, it is vital to carry out activities to reduce the risk of functional dependence as a health problem in the geriatric population [39]. Regular physical activity leads to a better quality of life in older people of different social contexts and economic incomes, being an effective therapy available to all in both health and community environments [39].

The limitations of the present study include: (i) the selection of the sample (intentional non-probabilistic) that only allowed for an association analysis; (ii) the geographical context of older women, which did not allow for the extrapolation of the results to other realities; (iii) not analyzing eating habits or sleep quality variables, which may have influenced the results of cardiometabolic risk; (iv) categorizing the physical activity habits according to the time of practice, rather than by more direct methods such as accelerometry. Among the strengths are: (i) a significant sample size (58%) of older women who participate in groups of older people in Osorno, Chile, which could provide crucial local contextual information; (ii) the analysis of blood fasting glucose, HDL cholesterol, and triglycerides; (iii) the simplicity of the body composition and physical performance assessments, which would allow their use and implementation in physical activity programs focused on older people; (iv) the analysis of the influence of physical activity on the analyzed variables. 

## 5. Conclusions

Physical inactivity was associated with increased hyperglycemia, hypertriglyceridemia, and high blood pressure, low HDL cholesterol levels, an excess body fat percentage, a lower fat-free mass, and a poorer HGS in dominant and non-dominant hands, TUG, and walking speed in Chilean older women. PA older women showed a lower cardiometabolic risk, better body composition, and better physical performance than PI older women. Public and private institutions should promote and support physical activity programs for older people to maintain and/or improve their health status.

## Figures and Tables

**Figure 1 ijerph-20-06688-f001:**
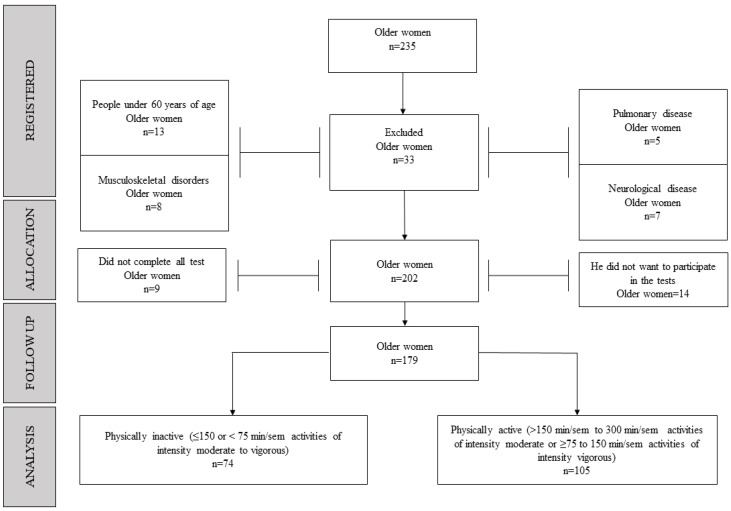
Flowchart of the recruitment process.

**Figure 2 ijerph-20-06688-f002:**
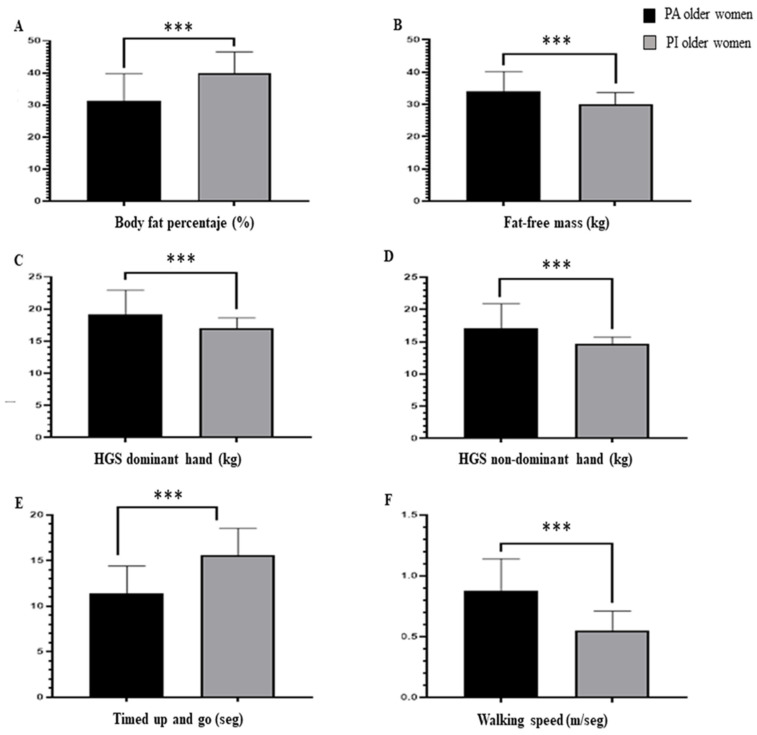
Comparison between physically active older women and physically inactive older women in terms of cardiometabolic variables. Abbreviations: SBP: systolic blood pressure. DBP: diastolic blood pressure. HDL: high-density lipoproteins. *** *p* < 0.001.

**Figure 3 ijerph-20-06688-f003:**
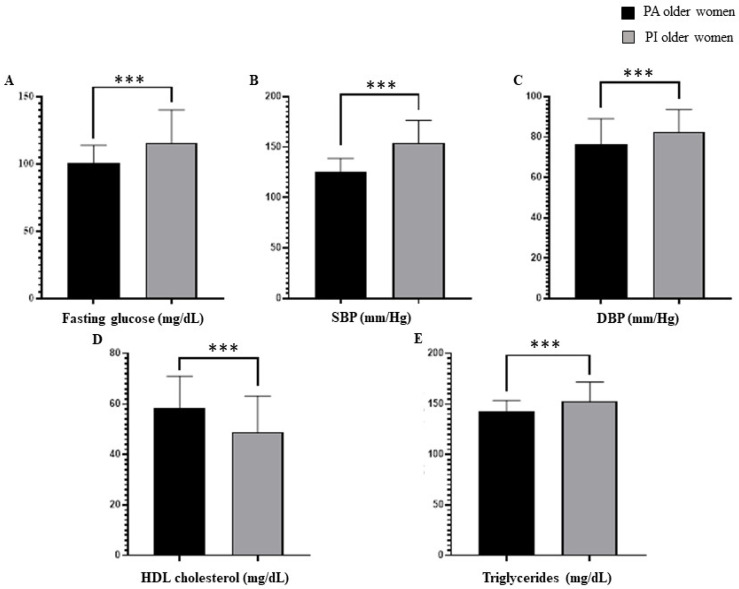
Comparison between physically active older women and physically inactive older women in terms of body composition and physical performance. Abbreviations: HGS: handgrip strength. TUG: timed up-and-go. *** *p* < 0.001.

**Figure 4 ijerph-20-06688-f004:**
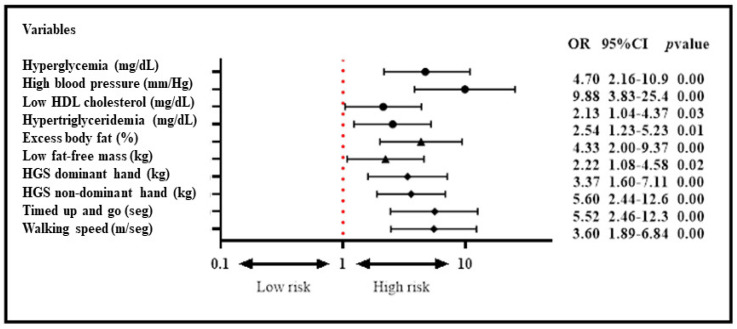
Association between physically inactive older women and cardiometabolic variables, body composition, and physical performance. Abbreviations: OR: odds ratio. CI: confidence interval. HDL: high-density lipoproteins.

**Table 1 ijerph-20-06688-t001:** Anthropometric characteristics and activities performed according to physical activity habits.

Variables	PA Older Women (*n* = 105)	PI Older Women (*n* = 74)
Age (years)	73.3 ± 2.5	77.5 ± 6.5
Body weight (kg)	60.3 ± 3.1	63.4 ± 4.1
Height (cm)	156.1 ± 4.3	152.3 ± 3.8
Body mass index (kg/m^2^)	24.8 ± 3.7	27.4 ± 3.9
Types of activities performed	Strength, balance, aerobic exercises, and Latin dance activities.	Sedentary activities including board games such as ludo, chess, checkers, dominoes, and cards.

PA: physically active. PI: physically inactive.

**Table 2 ijerph-20-06688-t002:** Characteristics of the sample according to physical activity habits.

Variables	PA Older Women (*n* = 105)	PI Older Women (*n* = 74)	*p*-Value
Fasting glucose (mg/dL)	100.4 ± 13.5	115.8 ± 24.4	*p* = 0.000
SBP (mmHg)	125.7 ± 13.1	154.2 ± 22.4	*p* = 0.000
DBP (mmHg)	76.6 ± 12.5	82.6 ± 11.0	*p* = 0.000
HDL cholesterol (mg/dL)	58.5 ± 12.6	49.0 ± 14.2	*p* = 0.000
Triglycerides (mg/dL)	142.2 ± 11.3	152.9 ± 18.9	*p* = 0.000
Body fat percentage (%)	31.3 ± 8.43	40.0 ± 6.59	*p* = 0.000
Fat-free mass (kg)	34.0 ± 6.16	30.1 ± 3.59	*p* < 0.001
HGS dominant hand (kg)	19.1 ± 3.85	17.0 ± 1.63	*p* = 0.000
HGS non-dominant hand (kg)	17.1 ± 3.78	14.6 ± 1.12	*p* < 0.001
TUG (seg)	11.4 ± 3.00	15.6 ± 2.93	*p* = 0.000
Walking speed (m/seg)	0.88 ± 0.26	0.55 ± 0.16	*p* < 0.001

PA: physically active. PI: physically inactive. SBP: systolic blood pressure. DBP: diastolic blood pressure. HDL: high-density lipoproteins. HGS: handgrip strength. TUG: timed up-and-go.

## Data Availability

The datasets generated during and/or analyzed during the current research are available from the Corresponding author upon reasonable request.

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
