# Peer review of "Association between Physical Activity Habits with Cardiometabolic Variables, Body Composition, and Physical Performance in Chilean Older Women"

_ijerph, 2023, doi:10.3390/ijerph20176688_

Round 1

Reviewer 1 Report

This is an interesting study carried out in the south of Chile, they associated cardiometabolic health with physical inactivity. In general, it is well-written, and add more evidence for the Chilean population. However, some revisions are needed.

Minor

Results of table 1 and figures 2 and 3 are the same; maybe table 1 should include sociodemographic characteristics if they were taken. 

The logistic regression is not clear in methods, just PI was outcome? what were the results for PA? or you compare PI women with PA? I suggest include in methods.  Do they include any covariates as confounders or were they mutually adjusted? 

line 44 lacks a quotation mark (") after Mayor

line 92, change "cross-sectional, descriptive, and comparative study with a quantitative approach" for analytical cross-sectional study

line 105, why describe 310, but the figure starts in 235? please clarify 

Author Response

As an attachment.

Reviewer 2 Report

In their work entitled “Association between physical activity habits with cardiometabolic variables, body composition and physical performance in Chilean older women” Hernandez-Martinez and colleagues evaluated the relationship between physical activity patterns and cardiometabolic measure, body composition and various measures of physical performance.

Comments:

1.      The introduction would be significantly improved with linguistic editing. Currently, it is unclear specifically what is being conveyed in some sections.

2.      Figures 2 and 3 the associated p values appear to only be shown in text and not in any table or figure. It would be beneficial to show the p values either in Figures 2 and 3 or ideally in Table 1 above.

3.      Were the p values in lines 211-212, 219-221, 228-229, 223, 238-239 exactly p=0.000 or p<0.001? If not an exact p=0.000, please indicate such in the manuscript.

4.      The power calculation is presented in a manner that may benefit from clarification.

The introduction would be significantly improved with linguistic editing. Currently, it is unclear specifically what is being conveyed in some sections.

Author Response

As an attachment.

Reviewer 3 Report

This manuscript addressed a really nice and important topic, however, there are some issues before publication.

1.     The author measured hand grip strength and some sarcopenia-related factors. However, this outcome is not mentioned in the manuscript. In the title, instead of body composition, it would be better to mention sarcopenia and also in the method and result section provide this information.

2.     More information regarding the type of PA should be presented in the result section.

3.     It should be better to divide the participants according to their level of PA based on MET categories to show a dose-response relationship.

4.      There are few studies addressing the importance of PA in older adults. The author may discuss the importance of PA for older adults by presenting the results of more studies in the discussion. Maybe presenting rehabilitative strategies is also helpful. For example, the following paper addressed this issue very well:

Rahmati M, Keshvari M, Koyanagi A, Yon DK, Lee SW, Shin JI, Smith L. The effectiveness of community ageing in place, advancing better living for elders as a biobehavioural environmental approach for disability among low-income older adults: a systematic review and meta-analysis. Age and Ageing. 2023 Apr 1;52(4):afad053.

The quality of English language should be improved.

Author Response

As an attachment.

Round 2

Reviewer 3 Report

Congratulations to the authors for the work done in the revised version.